# Nitric Acid Leaching for Magnesium Extraction from Asbestos Ore Waste: From DoE to Predictive Modeling and Cost-Efficient Optimization

**DOI:** 10.3390/molecules30224396

**Published:** 2025-11-13

**Authors:** Nikolay S. Ivanov, Oleg S. Kholkin, Arlan Z. Abilmagzhanov, Iskander E. Adelbayev, Sergey K. Oparin, Nataliya Ivanova, Vladislav Kudryashov

**Affiliations:** 1Laboratory of Applied Research, D.V. Sokolskiy Institute of Fuel, Catalysis and Electrochemistry JSC, Almaty 050010, Kazakhstan; n.ivanov@ifce.kz (N.S.I.); g.freeman-17@mail.ru (O.S.K.); iskanderadelbayev@gmail.com (I.E.A.); sergey_oparin@vk.com (S.K.O.); ivanovana.1989@outlook.com (N.I.); 2Renewable Energy Lab, National Laboratory Astana, Nazarbayev University, Astana 010000, Kazakhstan

**Keywords:** asbestos, leaching, mathematical planning, magnesium oxide, ecology, modeling, cost-efficiency optimization

## Abstract

Kazakhstan’s asbestos industry produces over 3 million tons of waste annually. The primary component of asbestos ore waste (AOW) is magnesium rich minerals. In this study, the extraction of magnesium from AOW with nitric acid (HNO_3_) was for the first time systematically studied. A series of experiments were conducted to optimize acid concentration (300–600 g/L), leaching temperature (55–95 °C), leaching time (60–180 min), solid-to-liquid ratio (1:3–1:7), and particle size, with the overall goal of maximizing magnesium extraction and cost efficiency. Our results provide dependence of magnesium extraction in the order of acid concentration  >  temperature  >  time  >  solid-to-liquid ratio, while particle size was found to be negligible. The cost-efficiency optimization demonstrated the positive impact of the relatively low acid concentrations (< 450 g/L) and temperatures between 65 and 85 °C, while the Protodyakonov model validated a linear dependence of the extraction rate on temperature and acid concentration. Our model demonstrates that extraction efficiencies of up to 90% can be achieved while reducing reagent use and lowering the overall cost of magnesium production. Leaching of magnesium by HNO_3_ also opens a pathway to a closed-cycle process, due to the formation of magnesium nitrate. The thermal decomposition of Mg(NO_3_)_2_ provides valuable products such as MgO and NO_2_ reused in HNO_3_ regeneration for subsequent cycles. The proposed model predicts magnesium extraction from asbestos ore depending on leaching parameters with reasonable accuracy.

## 1. Introduction

The Zhitikara chrysotile asbestos deposit, one of the largest reserves of its kind in the world, is located in Kazakhstan. Kostanay Minerals JSC, which develops the deposit, produces about 250,000 tons of asbestos fiber annually, 98% of which is exported. This level of production generates more than 3 million tons of tailings [1]. Approximately 20% of these tailings are processed into crushed stone and sand for sale, while the remainder is stored. In addition to the environmental challenges associated with waste accumulation, there is also significant commercial interest in recycling this material, as it contains 25–27% magnesium [2,3,4].

Magnesium belongs to the group of strategically significant elements [5,6], and its demand is increasing due to broader use in advanced high-tech industries. [7] The relatively low melting temperature of magnesium (650 °C), hexagonal close-packed crystal lattice, low density (1.74 g/cm^3^) and high specific strength determine the interest in it as a structural material of a new generation [8,9]. The high negative standard electrode potential of magnesium (–2.37 V), combined with the two-electron oxidation mechanism, determines its prospects for use in the field of energy conversion and storage [10], providing a theoretical specific capacity of 2205 mA·h/g [11], as an alternative to lithium-ion batteries. Magnesium compounds are used in various sectors of the modern economy. There are a number of the most widely used magnesium compounds: magnesium oxide, magnesium nitrate, magnesium chloride, etc. Magnesium oxide is used in the microelectronics industry [12], in medicine [13,14,15], as a refractory ceramic material [16]. Magnesium nitrate is used as a fertilizer [17] and even as a catalyst in the extraction of rubidium [18]. Special attention should also be paid to organomagnesium compounds. Magnesium is an important microelement for humans. A deficiency of this microelement causes the disease hypomagnesemia [19]. Its bioavailability depends on the form of the compound in which it enters the body. The most bioavailable forms are organic forms: citrate, glycinate, taurate, etc. [20]. Thus, magnesium is defined as an element of exceptional importance for human life and the development of modern technologies.

The largest share of the market in monetary terms is accounted for by magnesium oxide [21], which is used in a wide range of technological sectors. It is mainly obtained from magnesite (MgCO_3_), dolomite (CaCO_3_·MgCO_3_), talc (Mg_3_Si_4_O_10_(OH)_2_), etc. In industrial production, various processes are used for extraction from these minerals, depending on the type of raw material. But in a simplified way, they can be described as follows: treatment of the mineral with acid (H_2_SO_4_ or HCl) followed by purification from calcium impurities and calcination of magnesium salts into commercial magnesium oxide [22,23]. Researchers believe that this process is characterized by some problems: corrosion control, leaching rate, thermal energy costs, and the presence of Fe, Ni, and Cu impurities that reduce the process rate and require optimization [24].

Due to the depletion of classical magnesium minerals, there is another promising magnesium-containing mineral—chrysotile asbestos production tailings, containing about 36 wt.% MgO. Chrysotile (Mg_3_(Si_2_O_5_)(OH)_4_), the main mineral containing Mg^2+^ in these tailings, is a layered silicate. The regions of chrysotile asbestos mining are mainly located in Russia, China, Canada, Italy and Kazakhstan [25], where for several decades extensive mining and processing have produced hundreds of millions of tons of chrysotile asbestos tailings. There is concern that these open asbestos quarries are often left uncovered. Residual asbestos fibers in tailings storage facilities are subject to wind dispersal, which can cause diseases in the local population, such as asbestosis or lung cancer [26]. According to researchers’ estimates [27], about 232,000 deaths are registered annually worldwide. Consequently, the processing of chrysotile asbestos tailings with magnesium extraction can create an additional profitable product in the market and reduce the environmental burden of the accumulation of solid potentially hazardous waste in the region.

The main modern methods for extracting Mg^2+^ from silicate minerals can be distinguished and include either high-temperature roasting or acid leaching. In thermal treatment [3,28], chrysotile asbestos tailings are usually sintered with ammonium sulfate in a mass ratio of 1:1.2–1:1.5 at temperatures of 600–700 °C for 1–2 h. To increase the yield of magnesium, various additives are additionally introduced [24]. The average yield of magnesium under such conditions is 60–70%. A large number of works [29,30,31,32] are devoted to leaching with solutions of strong inorganic acids, mainly sulfuric and hydrochloric acids. Their use leads to the destruction of the mineral and the transfer of Mg^2+^ ions into solution already at temperatures of 20–25 °C. For example, in HCl the proportion of magnesium extracted from a ground asbestos sample in [30] increases linearly with the square root of time to about 65% extraction. In the case of treatment with sulfuric acid at moderate concentrations (30–60%), extraction of up to 84% Mg is achieved. Among the strong leaching agents, there is another widely available and relatively inexpensive one—nitric acid, the use of which in industry is underestimated.

Nitric acid leaching has advantages over other methods of asbestos processing: the simplicity of converting magnesium nitrate into the most profitable product, MgO [32]:(1)$$2Mg(NO_{3}{)}_{2}=2MgO+4NO_{2}+O_{2}$$

This approach is promising due to the prospect of using it in high-tech industries where high-purity substances uncontaminated by sulfate or chloride ions are required. As well as with the possibility of regenerating nitric acid during the technological process and creating a closed-loop raw material processing cycle [33]. In this regard, the aim of the present study is to investigate the process of leaching magnesium from asbestos production waste of the Zhitikara deposit with nitric acid using mathematical modeling methods to find the optimal conditions of the process.

Thus, the processing of magnesium-containing asbestos production waste reduces environmental costs and helps to obtain a profitable product in demand on the world market. The organization of modern production of this type will provide not only new jobs and additional tax revenues to the state budget but will also form a solid foundation for the development of high-tech innovative industries in demand on a global scale. This will also be consistent with the goals of sustainable development of the modern resource-saving economy.

## 2. Results

AOW is a fine powder of baked milk color which is waste from asbestos production. Figure 1 represents the elemental and phase analysis results of the AOW. Asbestos waste mainly consists of chrysotile (85%), accompanied by minor phases identified in the phase composition diagram. X-ray diffraction (XRD) patterns of the AOW indicates that the primary phase includes chrysotile Mg_3_Si_2_O_5_(OH)_4_, clinohumite (Mg, Fe)_9_Si_4_O_16_(F,OH)_4_, brucite Mg(OH)_2_, magnesium ferrite MgFe_2_O_4_, magnetite FeO·Fe_2_O_3_, and periclase MgO. SEM observation shows that asbestos waste consists of two fractions.

One is a fibrous structure, with fibers extending to several millimeters in length and diameters ranging from ~200 nm to the micron scale, surrounding smaller particles. Elemental and X-ray diffraction analyses indicate that these particles are mainly magnetite and magnesium ferrite, with sizes between 50 µm and 1 mm. After separation, it is evident that the same fibers are also present in much smaller dimensions, down to about 150 nm in diameter. The composition of the asbestos waste phase (magnetic and non−magnetic) after magnetic separation is presented in Figure 2.

The presence of multiple peaks in the diffraction patterns indicates the initial multicomponent composition of the asbestos waste before magnetic separation. The main components of the magnetic phase are periclase MgO, magnesium ferrite MgFe_2_O_4_, and magnetite FeO·Fe_2_O_3_ with magnetic properties. The corresponding diffraction pattern shows that the diffraction peaks near 22°, 35°, 40°, 63°, 68° and 90° (blue squares) correspond to the magnesium ferrite (111), (220), (311), (422), (511) and (533) planes of the cubic lattice (normal spinel-type), respectively. The diffraction peaks near 40°, 50°, 62° and 67° (red circles) are associated with magnetite (311), (400), (422) and (511) planes of the face-centered cubic lattice, respectively. Magnesium in the composition is generally present in the periclase form with corresponding diffraction peaks near 43°, 50°, and 75° (green rhombus) according to (111), (200), and (220) planes, respectively. The main components of the non-magnetic phase are asbestos complex compounds, including chrysotile Mg_3_Si_2_O_5_(OH)_4_, which is the commonly encountered form of asbestos, clinohumite (Mg, Fe)_9_Si_4_O_16_(F,OH)_4_, and brucite Mg(OH)_2_. The corresponding diffraction pattern shows that the diffraction peaks near 14°, 21°, 28°, 71°, and 86° (pink triangles) correspond to the chrysotile (200), (111), (004), (513), and (803) lattice planes, respectively. The diffraction peaks near 26° and 53° (orange pentagons) are associated with clinohumite (210), and (−122) lattice planes, respectively. Magnesium in the composition is generally present in the brucite form with corresponding diffraction peaks near 45° and 60° (brown stars) according to (101), (102) lattice planes, respectively. Diffuse peaks in the region of 20–25° and 40–45° for both magnetic and non-magnetic phases are associated with the presence of amorphous quartz particles.

The presented Mössbauer spectra also confirm the compositions of the iron-based compounds the separated phases. The Mössbauer spectrum of the non-magnetic part is a superposition of paramagnetic components (three doublets). Doublets 2 and 3 have parameters close to those of iron in the silicate minerals serpentine and greenalite Fe_3_Si_2_O_5_(OH)_4_, which contain Fe^3+^ and Fe^2+^ ions in their composition [34]. The shift of doublets is associated with partial replacement of Fe^2+^ by Mg^2+^ in the composition Fe_2_MgSi_2_O_5_(OH)_4_ [35], as well as the presence of magnesium minerals such as chrysotile Mg_3_Si_2_O_5_(OH)_4_. Doublet 1 has parameters close to those of iron hydroxide γ—FeOOH (lepidocrocite). Detailed analysis of the Mössbauer spectrum for the non-magnetic fraction reveals two magnetically ordered components accompanied by a weak paramagnetic contribution. The tetrahedral sublattice is characterized by an isomer shift of 0.26 mm/s, quadrupole splitting of 0.00 mm/s, an effective magnetic field of 489 kOe, and a relative spectral area of 5%, and the octahedral sublattice is characterized by an isomer shift of 0.67 mm/s, quadrupole splitting of 0.00 mm/s, an effective magnetic field of 458 kOe, and a relative spectral area of 13%. The paramagnetic components are represented by three doublets. The first doublet has an isomer shift of 1.15 mm/s, quadrupole splitting of 2.75 mm/s, and a relative spectral area of 37%. The second doublet is characterized by an isomer shift of 0.34 mm/s, quadrupole splitting of 0.56 mm/s, and a relative spectral area of 37%. The third doublet shows an isomer shift of 0.44 mm/s, quadrupole splitting of 0.77 mm/s, and a relative spectral area of 8%. The spectrum of the magnetic part is a superposition of magnetically ordered and paramagnetic components (two sextets and one doublet). Two magnetically ordered sextets correspond to magnetite FeO·Fe_2_O_3_ or magnesium ferrite MgO·Fe_2_O_3_ with close parameters. The doublet has parameters close to those of iron hydroxide γ—FeOOH (lepidocrocite), with similar values for the non-magnetic part (doublet 1) The magnetically ordered states correspond to nearly stoichiometric magnetite (Fe_3_O_4_). The tetrahedral sublattice shows an isomer shift of 0.28 mm/s, quadrupole splitting of 0.01 mm/s, an effective magnetic field of 492 kOe, and a relative spectral area of 33%. The octahedral sublattice exhibits an isomer shift of 0.67 mm/s, quadrupole splitting of 0.01 mm/s, an effective magnetic field of 461 kOe, and a relative spectral area of 64%.

There are several studies where strong acids have been selected for magnesium extraction from asbestos ore (Table 1). Hydrochloric acid (HCl) [1] and sulfuric acid (H_2_SO_4_) [30] have been widely used as effective leaching agents. Other approaches include ore roasting followed by water leaching with ammonium sulfate ((NH_4_)_2_SO_4_) [3], as well as leaching with weaker organic acids such as oxalic acid (C_2_H_2_O_4_) [36].

Magnesium is recognized as a critical raw material by the European Union, which highlights the importance of developing sustainable methods for its extraction. A comparative analysis of existing leaching technologies demonstrates that different reagents provide distinct advantages and limitations [38]. Among them, nitric acid was selected as the most promising reagent because it enables a closed-cycle process with acid regeneration. During leaching, magnesium is dissolved in the form of magnesium nitrate, Mg(NO_3_)_2_, which decomposes at a relatively low temperature of about 290–450 °C into magnesium oxide (MgO) and gaseous nitrogen oxides (NO_2_) [35]. The released nitrogen oxides can be absorbed by water to regenerate nitric acid, as reported in the literature on the thermal treatment of magnesium nitrate [39]. Although the nitric acid route may provide slightly lower extraction yields under identical conditions, it offers a major advantage through reagent recovery and reduced waste generation. As shown in Table 1 the main magnesium extraction methods confirms that the nitric acid leaching process represents the most environmentally and economically viable approach within the principles of green chemistry, as it excludes the use of additional reagents and minimizes the formation of by-products.

To estimate the minimum amount of nitric acid required to leach the Mg from asbestos ore was calculated based on mineralogical and elemental compositions. Based on this calculation, the minimum acid concentration was defined. The leaching process is expected to proceed through the following reactions:Mg_8.42_Fe_0.58_Si_4_O_16_(OH)_2_ + 18HNO_3_ = 8.42Mg(NO_3_)_2_ + 0.58Fe(NO_3_)_2_ + 4SiO_2_ + 10H_2_O(2)Mg(OH)_2_ + 2HNO_3_ = Mg(NO_3_)_2_ + 2H_2_O(3)Mg_3_(Si_2_O_5_)(OH)_4_ + 6HNO_3_ = 3Mg(NO_3_)_2_ + 2SiO_2_ + 5H_2_O(4)Fe_3_O_4_ + 8HNO_3_ = 2Fe(NO_3_)_3_ + Fe(NO_3_)_2_ + 4H_2_O(5)(Fe_0.51_Mg_0.49_)(Cr_0.75_Al_0.25_)_2_O_4_ + 8HNO_3_ = 0.51 Fe(NO_3_)_2_ + 0.49Mg(NO_3_)_2_ + 1.5Cr(NO_3_)_3_ + 0.5Al(NO_3_)_3_ + 4H_2_O(6)MgFe_2_O_4_ + 8HNO_3_ = Mg(NO_3_)_2_ + 2Fe(NO_3_)_3_ + 4H_2_O(7)MgO + 2HNO_3_ = Mg(NO_3_)_2_ + H_2_O(8)

Based on these reactions to fully dissolve 15 g of AOW, 21 g of nitric acid are required.

Analysis of the graphs in Figure 3 shows that the nitric acid concentration directly influences the leaching process. This dependence is linear and increases over the entire studied range from 75.92 to 85.96%. A higher concentration in the liquid phase allows for interaction with a greater amount of raw material, which explains the observed trend.

When considering the influence of the solid-to-liquid ratio, it should be emphasized that low ratios (3:1 and 4:1) result in an insufficient amount of liquid phase (acid) to fully wet the processed material. Under these conditions, the mixture remains in a semi-dry state, which significantly hinders the leaching process and prevents it from reaching completion, as confirmed by the graph (Figure 4). Ratios of 5:1, 6:1, and 7:1 provide enough liquid to ensure complete wetting of the solid phase. Consequently, the extraction degree reaches its maximum value (approximately 82%) and remains unchanged with further increases in the volume of the liquid phase.

Temperature is one of the main factors influencing the rate of magnesium dissolution in nitric acid. With increasing temperature, the extraction degree of magnesium rises considerably, reaching about 63% higher values compared to 55 °C. This behavior agrees with reported studies on the hydrometallurgy of magnesium silicates, where leaching of serpentine in nitric acid becomes much faster at temperatures above 70 °C [40]. The improvement is mainly associated with the acceleration of reaction kinetics, since higher temperatures reduce the activation energy and promote the diffusion of reactants. Experimental results also support this tendency: when leaching magnesium from slag, an increase in temperature from 50 °C to 100 °C led to roughly a 27% improvement in magnesium recovery [41]. However, further temperature growth for process intensification is not practical because it requires more complex and corrosion-resistant equipment. At higher temperatures, excess pressure develops in the system, and nitric acid may undergo partial decomposition, which decreases the efficiency of leaching. Therefore, the optimal temperature range for nitric acid leaching of magnesium lies in the region where the reaction rate is maximized without the appearance of side effects.

The duration of the leaching process has a strong effect on the extraction degree of the target component. Increasing the contact time of the solid phase with the nitric acid solution leads to more complete dissolution. This is explained by the fact that with longer contact, the probability of collisions between solid particles and acid molecules increases, which ultimately results in a higher extraction degree. Experimental data show that the maximum extraction (90.46%) is achieved after three hours of treatment.

The particle size of the solid phase has practically no influence on the leaching degree. The magnesium yield when the particle size changes from 0.071 to 1.25 mm, it varies between 79.28 and 81.94%, which means the effect of degree of grinding on the leaching process is negligible.

To visualize the efficiency of magnesium extraction from asbestos ore by leaching with nitric acid, three parameters were used: temperature, nitric acid concentration, and leaching time. When constructing the experimental model, the particle size of the asbestos ore was considered. The authors show, during magnesium leaching at times longer than 60 min, the effect of particle size becomes negligible. The target value for magnesium extraction was set at 60%. In the multi-factor model, several possible factors could influence the magnesium yield. The experimental matrix included particle size, but in the calculations this factor was negligible. As shown in Figure 5, the extraction rate increases with temperature and acid concentration. It can be seen that time is a crucial parameter in the leaching process, since the extraction rate drops to 68.67%, which means that acid concentration and temperature alone do not allow the desired extraction rate to be achieved. To better illustrate how these three parameters influence the extraction rate, a 3D plot is presented, where the three axes represent acid concentration, temperature, and time, while the color represents the magnesium extraction rate.

To find the optimal conditions where all these parameters provide the best extraction rate, we propose cost-efficiency calculations. These calculations include corrosion losses from the reactor, the time spent on extraction, and the cost of all chemical reagents, as described in the methods section. As can be seen, the best cost efficiency is represented in green in the range of acid concentration below 500 g/L and temperature between 65–85 °C. It can also be observed in several regions at higher concentrations and temperatures, but the cost model does not include the degradation of nitric acid at higher concentrations. It is also well known that concentrated nitric acid decomposes and, in contact with minerals, can produce NO_2_, which increases the corrosion rate and generates hazardous gases.

The statistical validation of the regression analysis was carried out to confirm the reliability and predictive quality of the developed model. Two fundamental assumptions were confirmed. The first one concerns the normal distribution of residuals, and the second refers to the homoscedasticity, that is, the constant variance of errors. The residuals were calculated as the difference between the experimentally measured magnesium extraction and the values predicted by the generalized regression equation for all 25 experiments. The mean value of residuals was close to zero, which indicates that the model built by the least-squares method is unbiased and adequately represents the experimental data.

The regression model showed a strong statistical fit, with a determination coefficient (R^2^) of 0.976 and a root mean square error (RMSE) of 2.57 percentage points for the training dataset. Cross-validation with five data folds gave an average R^2^ of 0.732 ± 0.221 and an RMSE of 6.83 ± 2.25 percentage points. These results confirm that the model is stable and has a good ability to predict new data. The model includes linear, quadratic, and interaction terms, which describe both the main and combined effects of the studied factors. Centering of variables around their mean values ensured stable estimation of interaction coefficients and improved numerical convergence.

The analysis of the residual distribution showed that most deviations were moderate. In 23 out of 25 experiments, the absolute deviation did not exceed 6.7 percentage points. The largest deviation, 9.44%, was observed in experiment 24, where the predicted yield (78.11%) was lower than the experimental value (87.55%). This experiment was performed at moderate temperature (65 °C), high nitric acid concentration (525 g/L), and a solid-to-liquid ratio of 7:1. Such a deviation may be related to local mass-transfer limitations or small experimental imperfections near the boundary of the studied region, which are difficult to describe by a polynomial regression model.

The residuals were evenly distributed across the full range of predicted values and did not show any trend of increasing or decreasing variance. This observation confirms the homoscedasticity of the model and indicates that its predictive accuracy remains stable throughout the whole experimental space. The most significant interactions (*p* < 0.05) were found between acid concentration and temperature, and between acid concentration and the solid-to-liquid ratio. These interactions are consistent with the physical nature of the leaching process, where both parameters jointly affect the reaction rate and the overall degree of magnesium extraction.

The practical application of the generalized equation lies in the optimal optimization of the technological process through the saving of resources (material, time, energy). It can also be useful for the selection of equipment. Applied to the considered technology of nitric acid leaching of magnesium, for example, it is known that the selected material for the reactor is the least subject to corrosion at an acid concentration of 285 g/L. In this case, when choosing the ratio of liquid phase to solid, it is necessary to calculate so that the amount of acid is greater than the stoichiometric one. For the selected concentration, the L:S ratio will be 6.4:1. At the same time, we want the process not to last more than 3 h and for the degree of magnesium leaching to be at least 95%. Figure 6b shows all possible options under the specified conditions. In addition to the reactor material, these calculations are also applicable to the choice of its volume, pump capacity, and power of the energy system.

Based on model and experimental optimization, the phase composition of the leaching residue was analyzed. The solid product after leaching mainly consists of quartz with trace amounts of magnetite. The characteristic diffraction peaks near 35°, 40°, and 75° correspond to the (220), (311), and (440) planes of the face-centered cubic lattice of magnetite (Figure 7). This confirms that under optimal leaching conditions, most magnesium-bearing minerals are dissolved, while inert silica remains as the primary residue phase.

## 3. Materials and Methods

### 3.1. Materials

The AOW (Asbestos Ore Waste) sample used in this work was taken from Kostanay Minerals JSC, located in Zhitikara, Kazakhstan. Distilled water was employed for the leaching and subsequent filtration of the asbestos residue. Nitric acid (HNO_3_), purchased from Sigma-Aldrich (St. Louis, MO, USA), served as the leaching agent for magnesium extraction. The choice of nitric acid was justified by the fact that magnesium in the leachate is present in the nitrate form, which facilitates subsequent separation.

### 3.2. Metods

#### 3.2.1. Acid Leaching

The agitation leaching experiments were carried out in sealed reactor equipped with a gas-release valve. A schematic representation of the leaching process is shown in Figure 8. Prior to leaching, the AOW was heated in a muffle furnace at 100 °C overnight to remove any residual liquid phase. Afterward, the asbestos ore waste was ground in a laboratory disk grinder to obtain a uniform powder. The grinding process was carried out in two stages. The first was a rough stage with a gap of 0.35 mm and a duration of 3 min. The second was a fine stage with a gap of 0.18 mm and a duration of 3 min, with manual mixing every 2 min. The rotation speed was 875 rpm, and the batch load was 100 g. The temperature of the grinder housing was kept below 50 degrees Celsius to avoid moisture loss from crystal hydrates. After each batch, the disks were cleaned with compressed air and ethanol.

Magnetic separation was carried out using a laboratory roll-type separator with magnetic susceptibility in the range of 10^−6^ to 10^−7^ m^3^/kg. The working gap between the poles was 2 mm, the magnetic field induction 0.7 to 1.0 T, the roller rotation speed was 0.5 m/s, and the chute inclination angle was 20 degrees. The process was performed under dry conditions, and the material passed through the separator three times to ensure efficient separation. A sample weight of 15 g was used for each experiment. The reactor temperature was controlled by a LT-108A thermostat (LOIP, Saint Petersburg, Russia) that was connected to the leaching reactor. The reactor was placed on a LS-110 shaker (LOIP, Russian), and stirring was performed using the rolling method in the horizontal plane. The reactor volume was 200 mL, and the shaking speed was 200 RPM. After leaching, the samples were filtered and washed with distilled water until a neutral pH was reached. To estimate the magnesium extraction rate, ICP-MS measurements were performed on both the solution and the leaching residue. All necessary calibrations were carried out using certified calibration standards, with the calibration curve constructed over the relevant concentration range.

#### 3.2.2. Design of Experiments

The Latin square planning method was applied in this study, and the Protodyakonov equation was used to construct the experimental model [42]. This approach allows for the evaluation of how multiple factors affect the magnesium leaching rate and facilitates the identification of the interrelationships among them. The model can also be employed to predict the process outcomes when the operating conditions are modified.

Based on the obtained data, a second-order regression model was developed to describe the relationship between process parameters and magnesium extraction. The model includes linear, quadratic, and cross terms that reflect both the main and combined effects of the studied factors. All variables were centered relative to their mean values, which ensured stable estimation of the interactions.

This experiment aimed to establish how the magnesium extraction process functionally depends on five selected parameters. The complete experimental matrix is presented in Table A1 in Appendix A. The factors chosen for the leaching experiments were nitric acid concentration (AC), liquid-to-solid ratio (L:S), leaching temperature (T), leaching time (t), and particle size (PS).

#### 3.2.3. Model Description

To describe the dependence of the magnesium leaching process on the studied parameters, an orthogonal design with five factors was applied: nitric acid concentration (AC), solid-to-liquid ratio (SL), temperature (T), leaching time (t), and particle size (PS). The experimental results were processed using the least squares method to derive partial functions Yi for each factor. The values of Yi were obtained by averaging the extraction degrees over the corresponding matrix experiments.

The magnesium leaching process was described using a generalized mathematical model obtained from the statistically significant partial functions of nitric acid concentration, solid-to-liquid ratio, temperature, and leaching time. The particle size factor was excluded as statistically insignificant. The model is expressed as:(9)$$Y_{p}=\frac{Y_{acid}Y_{ratio}Y_{T}Y_{t}Y_{PS}}{Y_{a}^{x-1}}$$
where Y_acid_, Y_ratio_, Y_T_, Y_t_, and Y_PS_ the partial function of the nitric acid concentration, solid-to-liquid ratio, temperature and time of the leaching experiments, and particle size, correspondingly. Y_p_ is the generalized function, and Y_a_ represents the overall mean of all considered values of the generalized function.

Partial functions were calculated by averaging the magnesium extraction degree for each factor level. For example, Y_acid_ at the first level of nitric acid concentration (300 g/L) was obtained by averaging experiments 1–5, and at the second level (375 g/L) by averaging experiments 11–15. (see Table A1 in the Appendix A) The same approach was applied to all other factors, and the overall mean value was 80.65.

As part of the multifactor analysis, the contribution of each variable was evaluated using multiple correlation coefficients and their statistical significance (see Table A2, Appendix A). For the parameter “particle size,” the nonlinear multiple correlation coefficient was R = 0.07 with a corresponding t_r_ value of 0.11. Since the t_r_ value is far below 2, this factor can be considered statistically insignificant.

#### 3.2.4. Calculation of the Cost Efficiency

The cost efficiency of the magnesium extraction process was calculated by estimating the total cost (C_total_). Several factors were taken into account, with the main components being the price of acid, energy consumption, and the time required for equipment operation and labor. In addition, we also considered the effect of corrosion losses, the solid-to-liquid ratio, and the amount of acid that remains after the leaching process, since this unreacted acid can be reused and therefore decreases the final cost of magnesium extraction. Altogether, the total cost was expressed as:(10)$$C_{total}=C_{acid}+C_{energy}+C_{time}+C_{corr}+C_{S:L}-C_{residue}$$

The C_acid_ term was calculated according to the market price of nitric acid per liter used in each experiment. The C_time_ component represented the operation and labor time, which was considered as a resource equivalent to monetary cost.

The energy cost C_energy_ was calculated as the sum of the heating energy and the holding energy. For the heating energy, the total energy was estimated by the formula:(11)$$Q=\sum_{i}m_{i}c_{p,i}∆T$$
where m_i_ is the total mass of the leaching solution, c_pi_ is the specific heat capacity depending on nitric acid concentration, and ∆T is the temperature difference.

For the holding part, we applied the heat transfer equation:(12)$$E_{hold}=UA∆Tt$$
where U is the overall heat transfer coefficient (taken as 10 W/m^2^K for the closed leaching reactor), A is the surface area of the leaching reactor, ∆T is the temperature difference, and t is the time the solution was kept at the set temperature. The total thermal energy therefore equals:(13)$$E_{total\;energy}=E_{heat\;up}+E_{hold}$$

The obtained energy values were converted into cost using the electricity price per kilowatt-hour (kWh).

The corrosion cost (C_corr_) was calculated as part of the total cost model to include material degradation during nitric acid leaching. The estimation was performed using a simplified engineering method suitable for process cost calculations. The corrosion cost C_corr_ was considered based on data for 13% Cr steel. It has been shown in [43] that the corrosion rate does not exhibit a simple linear dependence. For each leaching condition, the corrosion rate value was directly applied and multiplied by a proportional coefficient (K_corr_) to obtain the corresponding cost term:(14)$$C_{corr}=r_{corr}K_{corr}$$

This approximation allowed the effect of acid concentration on corrosion-related losses to be included in the economic model without requiring a separate materials study.

The solid-to-liquid ratio C_S:L_ also showed a strong influence on the process. At very low ratios the leachate becomes wax-like, which is not effective. On the other hand, when the ratio goes above 1:7, the cost increases because much larger volume of medium is required. Therefore, the S:L ratio should be carefully optimized to balance efficiency and cost.

Finally, after each leaching experiment, we collected the unreacted nitric acid, which can be reused in the following runs. This reuse directly decreases the total price of magnesium extraction and improves the overall cost efficiency of the process.

#### 3.2.5. Characterization

The microstructures and surface elemental composition of the samples were analyzed by scanning electron microscopy (JSM 6610 LV, JEOL Ltd., Tokyo, Japan) and energy dispersive spectrometry (INCA Energy 450, Oxford Instruments, Abingdon, UK). X-ray diffraction analysis was carried out using an automated diffractometer (DRON-3, Burevestnik, St. Peterburg, Russia) with a CuKα-emission spectrum, β-filter. The conditions for recording diffractograms were as follows: U = 35 kV; I = 20 mA; and scan rate 2 grad/min. Mössbauer spectroscopy was used to study iron-containing phases. A Mössbauer spectrometer MS-1104Em (Research Institute of Physics, Rostov–on–Don, Russia) was used: source Co-57 in Rh matrix; activity—100 mCi. The spectra were processed using the least squares method. Isomeric shift (IS) values are given relative to α-Fe. All measurements were collected at 293 K. The imaging mode was ‘at lumen’. The amount of magnesium extracted was quantified via inductively coupled plasma mass spectrometry (ICP-MS, Agilent Technologies, Santa Clara, CA, USA).

## 4. Conclusions

This study demonstrates that nitric acid is a stable and effective leaching agent for extracting magnesium from waste asbestos ore at the Zhetikara deposit. A mathematical model built on experimental data predicts the magnesium extraction with high accuracy (R^2^ 0.94, standard error 4.39%). The model highlights temperature and acid concentration as the key factors influencing magnesium recovery. The results show that acid concentration has the strongest effect on the extraction, followed by temperature, then leaching time, and the solid-to-liquid ratio. In contrast, particle size has only a minor impact on the extraction yield. The model also suggests optimal conditions for magnesium recovery, the nitric acid concentration around 450 g/L and a temperature between 65 °C and 85 °C. Under these conditions, magnesium recovery can reach up to 90%.

Using nitric acid offers both high extraction efficiency and environmental advantages because the process operates in a closed chemical cycle. During leaching, magnesium nitrate (Mg(NO_3_)_2_) forms and can be thermally decomposed into magnesium oxide (MgO) and nitrogen oxides (NO_2_). These nitrogen oxides can be converted into nitric acid through absorption in water with partial oxidation of nitric oxide, which allows regeneration of nitric acid and reduction of chemical waste by up to 5%. The proposed nitric acid leaching route is part of a full cycle for recovering magnesium oxide from asbestos ore. It combines efficient magnesium extraction with acid regeneration and the production of valuable MgO. This approach ensures stable process performance, lower reagent consumption, and the sustainable use of industrial waste.

## Figures and Tables

**Figure 1 molecules-30-04396-f001:**
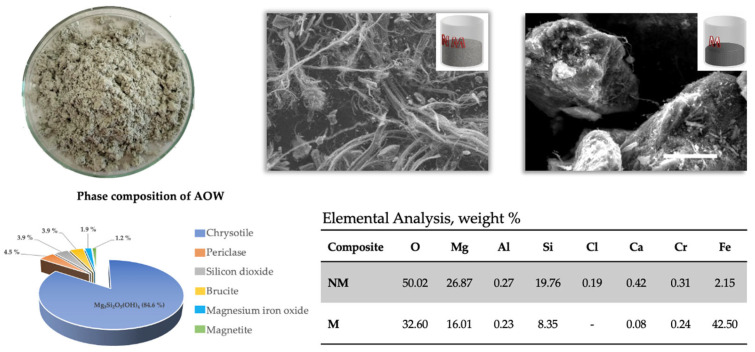
Optical micrograph and SEM images of asbestos waste, phase composition of asbestos ore and their elemental composition of NM and M samples.

**Figure 2 molecules-30-04396-f002:**
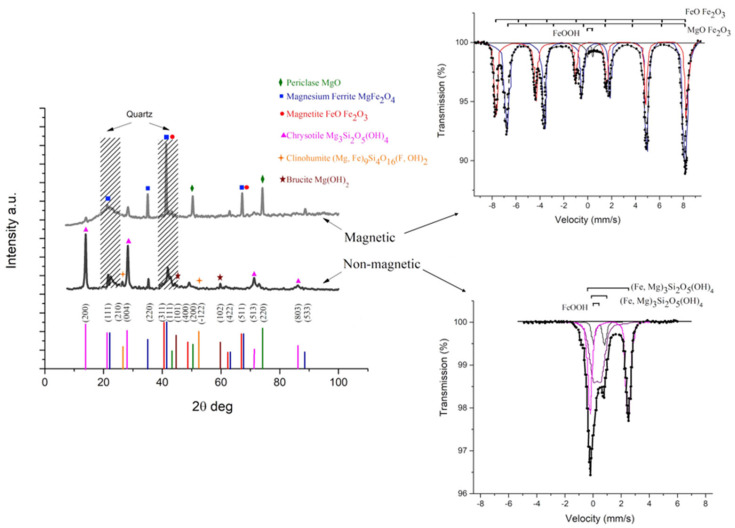
XRD patterns and Mössbauer spectra of the asbestos waste phases (magnetic and non-magnetic) after magnetic separation.

**Figure 3 molecules-30-04396-f003:**
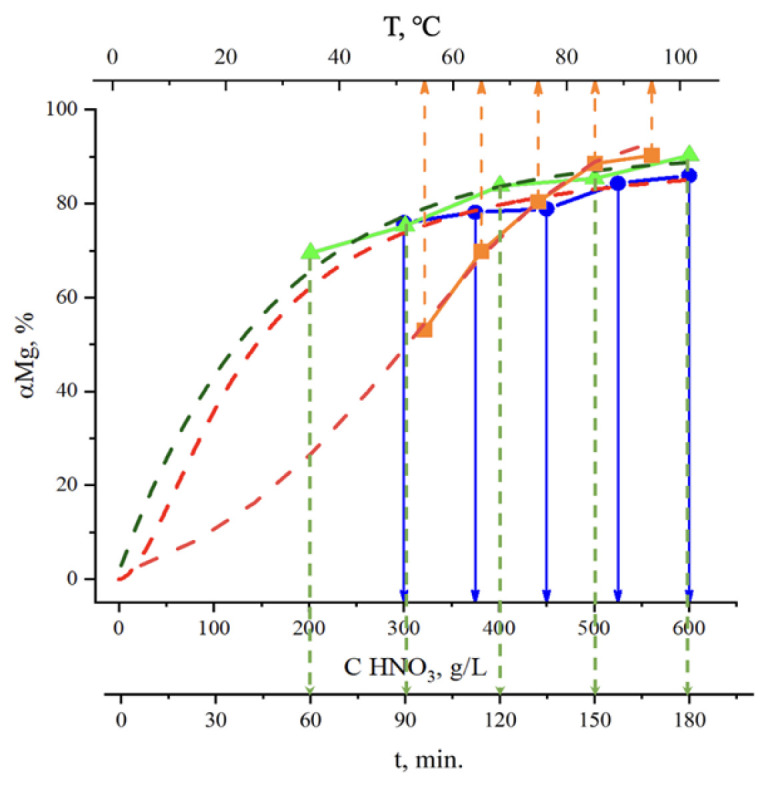
Effect of nitric acid concentration, temperature, and time on magnesium extraction. The dashed lines represent model–based approximations. The green triangles correspond to leaching time, the orange squares denote temperature, and the blue circles indicate nitric acid concentration. Vertical lines mark the respective x-axis values for each parameter.

**Figure 4 molecules-30-04396-f004:**
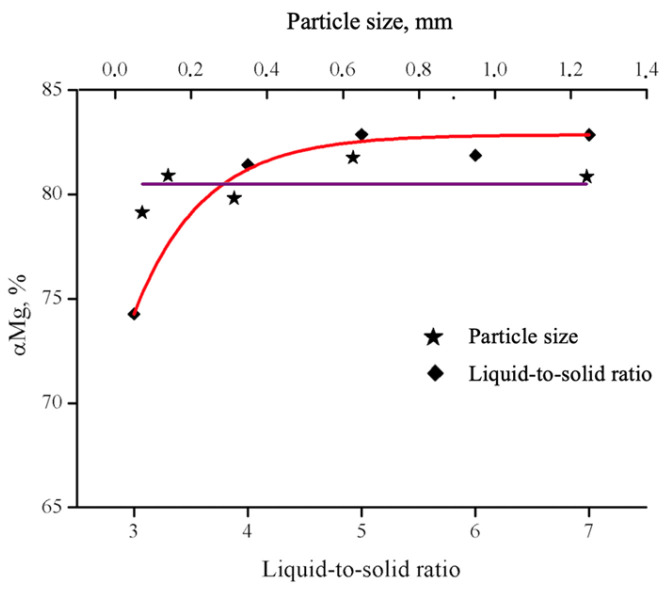
Magnesium extraction degree as a function of S:L ratio and PS particles size. Diamonds correspond to the liquid-to-solid ratio, while stars represent particle size.

**Figure 5 molecules-30-04396-f005:**
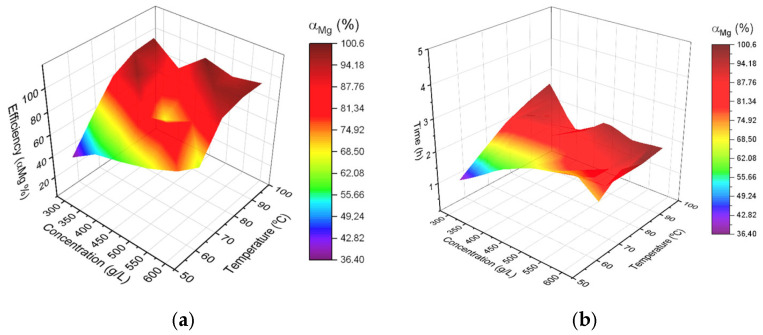
The 3D visualization of magnesium extraction depends on experimental setup, where (**a**) concentration, temperature and efficiency, and (**b**) the magnesium extraction is represented by color mapping depend of three parameters: acid concentration, leaching temperature and time.

**Figure 6 molecules-30-04396-f006:**
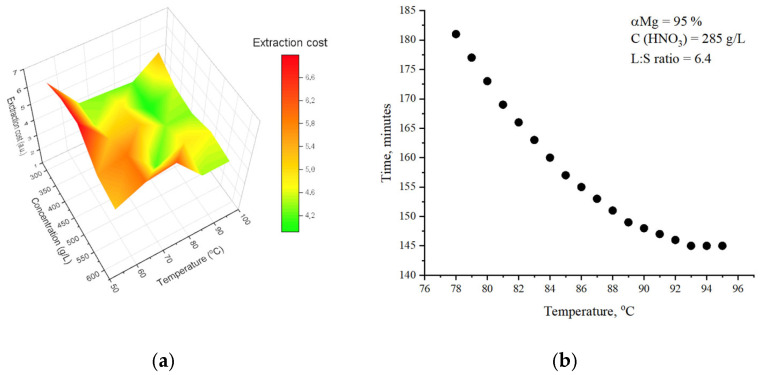
(**a**) The cost efficiency of magnesium extraction from AOW by nitric acid based on experimental data (**b**) calculated values of process parameters with default settings for magnesium extraction and acid concentration.

**Figure 7 molecules-30-04396-f007:**
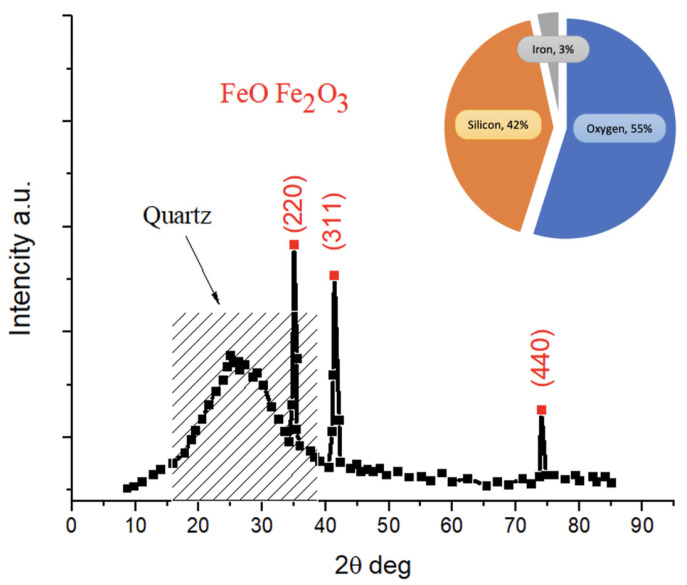
XRD data and elemental composition of silicon residue obtained after leaching of AOW by optimal parameters.

**Figure 8 molecules-30-04396-f008:**
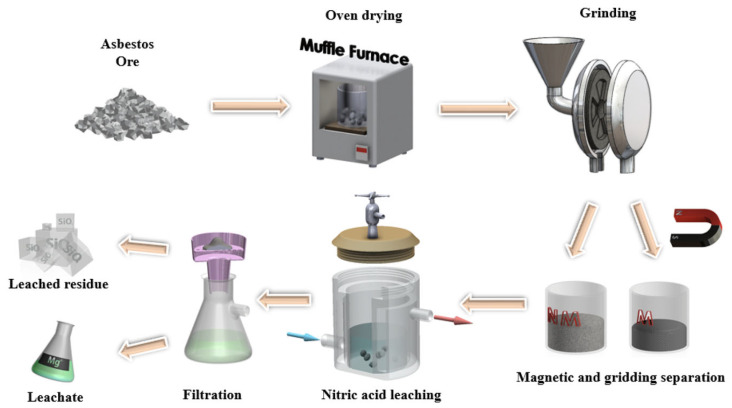
Schematic diagram of leaching experiment.

**Table 1 molecules-30-04396-t001:** Technologies for asbestos processing using various leaching agents.

№	Leaching Agent	Conditions	Advantages	Disadvantages	Ref.
	H_2_SO_4_	Serpentinite; C (H_2_SO_4_) = 10–60%; T = 90 °C; S/L = 1:10; t = 2 h.	Mg yield 85–90%; Low corrosive activity of the medium; Closed-cycle process.	Difficult purification and production of MgO; High decomposition temperature (700 °C); Difficult filtration of SiO_2_ precipitate.	[37]
	HCl	Chrysotile; C(HCl) = 2–6 M; T = 60–90 °C; t = 120 min.	Mg yield 89–92%; Production of pure MgO; Closed-cycle process; HCl regeneration; CO_2_-neutral technology.	High decomposition temperature (800 °C); High corrosive activity of HCl; Significant Fe^3+^ dissolution; Difficult filtration of SiO_2_ precipitate.	[28,30,31]
	(NH_4_)_2_SO_4_	Chrysotile tailings; (NH_4_)_2_SO_4_: solid = 1:1.15; T = 700–800 °C; t = 2 h; Aqueous leaching stage 60 °C: S/L = 1:10.	Mg yield up to 95%; closed-cycle process.	High decomposition temperature (800 °C); Large-scale process; Difficult purification and production of MgO; Absence of a closed cycle.	[3]
	HNO_3_	Chrysotile tailings; C(HNO_3_) = 450 g/L; T = 65–85 °C; t = 2–3 h.	Mg yield 90%; production of high-purity MgO; HNO_3_ regeneration; Moderate decomposition temperature (<500 °C); Closed-cycle process.	Formation of toxic nitrogen oxides (NO, NO_2_).	[present article]
	H_2_C_2_O_4_	Chrysotile asbestos; C(H_2_C_2_O_4_) = 0.05 M; t = 8 days; T = 26 °C.	Eco-friendly reagents; Moderate decomposition temperature (480 °C).	Mg yield 8.3%; Low leaching rate; Difficult purification of MgC_2_O_4_.	[38]

## Data Availability

Data will be made available upon request.

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
