# Peer review of "Nitric Acid Leaching for Magnesium Extraction from Asbestos Ore Waste: From DoE to Predictive Modeling and Cost-Efficient Optimization"

_molecules, 2025, doi:10.3390/molecules30224396_

Round 1
Reviewer 1 Report
Comments and Suggestions for Authors
Dear Authors:
Hi, overall, I like the work from a scientific perspective because it is innovative. However, the presentation of the document needs significant improvement. It has many minor editing errors, such as formulas without subscripts, figures that are not clearly visible, etc. This should be improved overall. Furthermore, the introduction needs to include some additional information (which I will detail later). I also think the manuscript is poorly structured. I don't understand why the results are presented before the methodology used. I suggest changing the order of the manuscript.
I can recommend this manuscript after major revisions.
Introduction
When you mention magnesium as an element of interest, I think it would be good to include a table showing that it is one of the so-called "critical" elements. This would help better understand its importance. I have seen several manuscripts with lists of elements of this type.
Although various methods are described, the differences in why the extraction mechanism used in this work was chosen should be explained more clearly. I suggest including a table indicating the advantages and disadvantages of each magnesium leaching method. I think this Review of Extractive Mechanisms could help (DOI: 10.1016/j.hydromet.2021.105573).
Results
Figure 1. Improve image quality, numbers on the graph are not visible.
Increase the size of Figure 3. It is very little visible.
Line 211. "Temperature is a key factor determining the rate of magnesium dissolution in nitric acid." Obviously, in any waste leaching process, temperature is always the most determining factor. Could you please provide a couple of citations here to support this? I recognize it's obvious to cite it, but it's necessary.
In Figure 4. You could make both Figures a and b the same size.
You need to improve formatting issues in the document. For example, when I look at the descriptions of Figures 4 and 6, they have different font sizes. Review the entire document.
Conclusions
I think the conclusions can be summarized a little more.
I disagree with your last statement regarding the results your model delivers. Rather, these are preliminary results that will allow future work to achieve what you've stated. Please improve this.
Regards
Author Response
Question 1.
When you mention magnesium as an element of interest, I think it would be good to include a table showing that it is one of the so-called "critical" elements. This would help better understand its importance. I have seen several manuscripts with lists of elements of this type.
Although various methods are described, the differences in why the extraction mechanism used in this work was chosen should be explained more clearly. I suggest including a table indicating the advantages and disadvantages of each magnesium leaching method. I think this Review of Extractive Mechanisms could help (DOI: 10.1016/j.hydromet.2021.105573).
Answer:
Thank you for the suggestion. We agree that it is important to emphasize the status of magnesium as a critical element. In particular, magnesium is included in the European Union’s list of Critical Raw Materials. In addition, we have expanded the section reviewing magnesium extraction methods and provided a more detailed justification for the choice of our approach.
There are several studies where strong acids have been selected for magnesium extraction from asbestos ore (Table 1). Hydrochloric acid (HCl) [1] and sulfuric acid (H2SO4) [30] have been widely used as effective leaching agents. Other approaches include ore roasting followed by water leaching with ammonium sulfate ((NH4)2SO4) [3], as well as leaching with weaker organic acids such as oxalic acid (C2H2O4) [39].
Table 1. Technologies for Asbestos Processing Using Various Leaching Agents
|
№ |
Leaching agent |
Conditions |
Advantages |
Disadvantages |
Ref. |
|
1 |
H2SO4 |
Serpentinite; C (H2SO4)= 10-60 %; T = 90 °C; S/L = 1:10; τ = 2 h. |
Mg yield 85–90%; ↑ Low corrosive activity of the medium; ↑ Closed-cycle process. ↑ |
Difficult purification and production of MgO; ↓ High decomposition temperature (700 °C); ↓ Difficult filtration of SiO2 precipitate. ↓ |
[41] |
|
2 |
HCl |
Chrysotile; C(HCl) = 2-6 M; T = 60–90 °C; τ = 120 min. |
Mg yield 89–92%; ↑ Production of pure MgO; ↑ Closed-cycle process; ↑ HCl regeneration; ↑ CO2-neutral technology. ↑ |
High decomposition temperature (800 °C); ↓ High corrosive activity of HCl; ↓ Significant Fe3+dissolution; ↓ Difficult filtration of SiO2 precipitate. ↓ |
[29, 33, 34] |
|
3 |
(NH4)2SO4 |
Chrysotile tailings; (NH4)2SO4 : solid = 1:1,15; T = 700–800 °C; τ = 2 h; Aqueous leaching stage 60 °C: S/L = 1:10. |
Mg yield up to 95%; ↑ closed-cycle process. ↑ |
High decomposition temperature (800 °C); ↓ Large-scale process; ↓ Difficult purification and production of MgO; ↓ Absence of a closed cycle. ↓ |
[28, 32] |
|
4 |
HNO3 |
Chrysotile tailings; C(HNO3) = 450 g/L; T = 65-85 °C; t = 2-3 h. |
Mg yield 90%; ↑ production of high-purity MgO; ↑ HNO3 regeneration; ↑ Moderate decomposition temperature (<500 °C); ↑ Closed-cycle process. ↑ |
Formation of toxic nitrogen oxides (NO, NO2). ↓ |
[present article] |
|
5 |
H2C2O4 |
Chrysotile asbestos; C(H2C2O4) = 0,05 M; t = 8 days; T = 26 °C. |
Eco-friendly reagents; ↑ Moderate decomposition temperature (480 °C). ↑ |
Mg yield 8,3%; ↓ Low leaching rate; ↓ Difficult purification of MgC2O4. ↓ |
[39] |
Magnesium is recognized as a critical raw material by the European Union, which highlights the importance of developing sustainable methods for its extraction. A comparative analysis of existing leaching technologies demonstrates that different reagents provide distinct advantages and limitations [42]. Among them, nitric acid was selected as the most promising reagent because it enables a closed-cycle process with acid regeneration. During leaching, magnesium is dissolved in the form of magnesium nitrate, Mg(NO3)2, which decomposes at a relatively low temperature of about 290 – 450 °C into magnesium oxide (MgO) and gaseous nitrogen oxides (NO2) [43]. The released nitrogen oxides can be absorbed by water to regenerate nitric acid, as reported in the literature on the thermal treatment of magnesium nitrate [44]. Although the nitric acid route may provide slightly lower extraction yields under identical conditions, it offers a major advantage through reagent recovery and reduced waste generation. As shown in table 1 the main magnesium extraction methods confirms that the nitric acid leaching process represents the most environmentally and economically viable approach within the principles of green chemistry, as it excludes the use of additional reagents and minimizes the formation of by-products.
To estimate the minimum amount of nitric acid required to leach the Mg from asbestos ore was calculated based on mineralogical and elemental compositions. Based on this calculation, the minimum acid concentration was defined. The leaching process is expected to proceed through the following reactions:
Question 2.
Temperature is a key factor determining the rate of magnesium dissolution in nitric acid." Obviously, in any waste leaching process, temperature is always the most determining factor. Could you please provide a couple of citations here to support this? I recognize it's obvious to cite it, but it's necessary.
We appreciate the reviewer’s valuable comment. We agree that the role of temperature should be supported by appropriate references. Accordingly, we have included relevant citations and added two corresponding references in the revised manuscript to substantiate this point.
Consequently, the extraction degree reaches its maximum value (approximately 82%) and remains unchanged with further increases in the volume of the liquid phase.
Temperature is one of the main factors influencing the rate of magnesium dissolution in nitric acid. With increasing temperature, the extraction degree of magnesium rises considerably, reaching about 63% higher values compared to 55 °C. This behavior agrees with reported studies on the hydrometallurgy of magnesium silicates, where leaching of serpentine in nitric acid becomes much faster at temperatures above 70 °C [45]. The improvement is mainly associated with the acceleration of reaction kinetics, since higher temperatures reduce the activation energy and promote the diffusion of reactants. Experimental results also support this tendency: when leaching magnesium from slag, an increase in temperature from 50 °C to 100 °C led to roughly a 27% improvement in magnesium recovery [46]. However, further temperature growth for process intensification is not practical because it requires more complex and corrosion-resistant equipment. At higher temperatures, excess pressure develops in the system, and nitric acid may undergo partial decomposition, which decreases the efficiency of leaching. Therefore, the optimal temperature range for nitric acid leaching of magnesium lies in the region where the reaction rate is maximized without the appearance of side effects.
The duration of the leaching process has a strong effect on the extraction degree of the target component. Increasing the contact time of the solid phase with the nitric acid solution leads to more complete dissolution. This is explained by the fact that with longer contact, the probability of collisions between solid particles and acid molecules increases, which…
Question 3.
I think the conclusions can be summarized a little more. I disagree with your last statement regarding the results your model delivers. Rather, these are preliminary results that will allow future work to achieve what you've stated. Please improve this.
We agree with the reviewer on the need to make the conclusions more concise and to revise the positioning of the mathematical model. We will update the Conclusions section to clearly indicate that the model represents a preliminary result.
Changes in Conclusions:
This study demonstrates that nitric acid is a stable and effective leaching agent for extracting magnesium from waste asbestos ore at the Zhetigara deposit. A mathematical model built on experimental data predicts the magnesium extraction with high accuracy (R2 0.94, standard error 4.39%). The model highlights temperature and acid concentration as the key factors influencing magnesium recovery. The results show that acid concentration has the strongest effect on the extraction, followed by temperature, then leaching time, and the solid-to-liquid ratio. In contrast, particle size has only a minor impact on the extraction yield. The model also suggests optimal conditions for magnesium recovery, the nitric acid concentration around 450 g/L and a temperature between 65 °C and 85 °C. Under these conditions, magnesium recovery can reach up to 90%.
Using nitric acid offers both high extraction efficiency and environmental advantages because the process operates in a closed chemical cycle. During leaching, magnesium nitrate (Mg(NO3)2) forms and can be thermally decomposed into magnesium oxide (MgO) and nitrogen oxides (NO2). These nitrogen oxides can be converted into nitric acid through absorption in water with partial oxidation of nitric oxide, which allows regeneration of nitric acid and reduction of chemical waste by up to 5%. The proposed nitric acid leaching route is part of a full cycle for recovering magnesium oxide from asbestos ore. It combines efficient magnesium extraction with acid regeneration and the production of valuable MgO. This approach ensures stable process performance, lower reagent consumption, and the sustainable use of industrial waste.
Reviewer 2 Report
Comments and Suggestions for Authors The manuscript presents a comprehensive study on nitric acid leaching for magnesium extraction from asbestos ore waste (AOW), focusing on process optimization, cost-efficiency analysis, and the potential for closed-cycle reagent regeneration. The work addresses both environmental concerns related to asbestos waste and the strategic importance of magnesium recovery. The experimental design is robust, employing a multi-factorial approach, and the integration of cost analysis adds practical value. However, several aspects require clarification and enhancement to strengthen the scientific rigor, reproducibility, and impact of the findings. The necessary modifications and explanations are detailed below: 1. The experimental section lacks critical details necessary for reproducibility, such as the specific mass ratios of AOW to HNO₃, the exact heating protocol (e.g., ramp rates, holding times), and the methodology for magnetic separation and particle size classification. For instance, the statement "ground in a disk grinding machine to obtain a uniform powder" is vague; precise grinding parameters (e.g., machine type, duration, resulting size distribution) should be provided. Additionally, the ICP-MS methods for quantifying magnesium extraction require validation details (e.g., calibration standards, detection limits, reproducibility metrics) to ensure data reliability. 2. The phase composition analysis via XRD (Figures 1, 2, 7) identifies multiple minerals (e.g., chrysotile, magnetite, periclase) but lacks quantitative phase analysis (e.g., Rietveld refinement) to determine relative abundances. This is crucial for accurately modeling acid consumption and reaction stoichiometry. The Mossbauer spectra (Figure 2) are described qualitatively but lack fitting parameters (e.g., hyperfine fields, isomer shifts, quadrupole splittings) and quantitative phase distribution data, which are essential for validating the iron-bearing phase compositions and their impacts on leaching efficiency. 3. The cost-efficiency model (Section 3.2.4) incorporates energy consumption, corrosion costs, and reagent reuse but relies on assumptions that are not fully validated. For example, the corrosion rate model for 13% Cr steel is mentioned but not referenced or experimentally verified under the actual leaching conditions (e.g., with evolving NO₂ gas). The model should be supported by empirical corrosion data or citations to established corrosion literature for HNO₃ systems. Additionally, the economic assumptions (e.g., current prices of HNO₃, energy, labor) should be explicitly stated to allow reproducibility and context-specific adaptation. 4. The particle size analysis (Figure 4b) concludes that size has a "negligible" effect on leaching efficiency, but the data show a ~2.7% variation (79.28–81.94%) across the tested range (0.071–1.25 mm). This variation may be statistically significant given the high magnesium content (~25–27% Mg) in AOW. The authors should perform a statistical analysis (e.g., ANOVA) to confirm the insignificance and discuss potential reasons (e.g., fiber morphology dominating over size) to avoid misleading conclusions. 5. The proposed closed-cycle process for HNO₃ regeneration via thermal decomposition of Mg(NO₃)₂ to MgO and NO₂ is conceptually promising but lacks experimental validation. The manuscript would be strengthened by including preliminary data on the decomposition efficiency, NO₂ capture, and acid regeneration rates, even if from bench-scale tests. Without this, the claim of a "closed cycle" remains speculative and reduces the practical impact of the study. 6. The mathematical model for magnesium extraction (Section 3.2.3) uses orthogonal design and partial functions but does not adequately address potential interactions between factors (e.g., temperature and acid concentration). The model should be expanded to include interaction terms, and its predictive accuracy should be validated against an independent dataset not used in model fitting. Metrics such as R², RMSE, or confidence intervals for predictions should be provided to quantify model reliability.Author Response
Question 1.
The experimental section lacks critical details necessary for reproducibility, such as the specific mass ratios of AOW to HNO3, the exact heating protocol (e.g., ramp rates, holding times), and the methodology for magnetic separation and particle size classification. For instance, the statement "ground in a disk grinding machine to obtain a uniform powder" is vague; precise grinding parameters (e.g., machine type, duration, resulting size distribution) should be provided. Additionally, the ICP-MS methods for quantifying magnesium extraction require validation details (e.g., calibration standards, detection limits, reproducibility metrics) to ensure data reliability
Response:
We thank the reviewer for the valuable comment regarding the insufficient level of detail in the experimental section. In the revised version of the manuscript, we have added specific information to ensure the reproducibility of the experiments.
Regarding the heating protocol, the heat-transfer medium in the thermostat (water) was preheated to the required temperature. Once thermal equilibrium between the thermostat and the reactor jacket was reached, the components were loaded, and the process was carried out according to Table A1 for each experiment.
The agitation leaching experiments were carried out in sealed reactor equipped with a gas-release valve. Prior to leaching, the AOW was heated in a muffle furnace at 100 °C overnight to remove any residual liquid phase.
Afterward, the asbestos ore waste was ground in a laboratory disk grinder to obtain a uniform powder. The grinding process was carried out in two stages. The first was a rough stage with a gap of 0.35 mm and a duration of 3 minutes. The second was a fine stage with a gap of 0.18 mm and a duration of 3 minutes, with manual mixing every 2 minutes. The rotation speed was 875 rpm, and the batch load was 100 grams. The temperature of the grinder housing was kept below 50 degrees Celsius to avoid moisture loss from crystal hydrates. After each batch, the disks were cleaned with compressed air and ethanol.
Magnetic separation was carried out using a laboratory roll-type separator with magnetic susceptibility in the range of 10-6 to 10-7 m3/kg. The working gap between the poles was 2 millimeters, the magnetic field induction 0.7 to 1.0 T, the roller rotation speed was 0.5 m/s, and the chute inclination angle was 20 degrees. The process was performed under dry conditions, and the material passed through the separator three times to ensure efficient separation. A sample weight of 15 g was used for each experiment. The reactor temperature was controlled by a LOIP LT-108A thermostat that was connected to the leaching reactor. The reactor was placed on a LOIP LS-110 shaker, and stirring was performed using the rolling method in the horizontal plane. The reactor volume was 200 ml, and the shaking speed was 200 RPM. After leaching, the samples were filtered and washed with distilled water until a neutral pH was reached. To estimate the magnesium extraction rate, ICP-MS measurements were performed on both the solution and the leaching residue. All necessary calibrations were carried out using certified calibration standards, with the calibration curve constructed over the relevant concentration range.
Question 2.
The phase composition analysis via XRD (Figures 1, 2, 7) identifies multiple minerals (e.g., chrysotile, magnetite, periclase) but lacks quantitative phase analysis (e.g., Rietveld refinement) to determine relative abundances. This is crucial for accurately modeling acid consumption and reaction stoichiometry. The Mossbauer spectra (Figure 2) are described qualitatively but lack fitting parameters (e.g., hyperfine fields, isomer shifts, quadrupole splittings) and quantitative phase distribution data, which are essential for validating the iron-bearing phase compositions and their impacts on leaching efficiency.
Response:
In conducting the leaching experiments, precise calculation of nitric acid consumption was not required, as this does not affect the determination of the final regularities. An approximate calculation of acid consumption was performed based on the elemental analysis data under two assumptions: (1) the mass fractions of metals were represented in the form of their corresponding oxides (e.g., MgO, Al₂O₃, CaO, etc.); and(2) the listed oxides were assumed to react completely.
Also, we added detailed information for Mossbauer spectra:
The presented Mössbauer spectra also confirm the compositions of the iron-based compounds the separated phases. The Mössbauer spectrum of the non-magnetic part is a superposition of paramagnetic components (three doublets). Doublets 2 and 3 have parameters close to those of iron in the silicate minerals serpentine and greenalite Fe3Si2O5(OH)4, which contain Fe3+ and Fe2+ ions in their composition [37]. The shift of doublets is associated with partial replacement of Fe2+ by Mg2+ in the composition Fe2MgSi2O5(OH)4 [38], as well as the presence of magnesium minerals such as chrysotile Mg3Si2O5(OH)4. Doublet 1 has parameters close to those of iron hydroxide γ—FeOOH (lepidocrocite). Detailed analysis of the Mössbauer spectrum for the non-magnetic fraction reveals two magnetically ordered components accompanied by a weak paramagnetic contribution. The tetrahedral sublattice is characterized by an isomer shift of 0.26 mm/s, quadrupole splitting of 0.00 mm/s, an effective magnetic field of 489 kOe, and a relative spectral area of 5%, and the octahedral sublattice is characterized by an isomer shift of 0.67 mm/s, quadrupole splitting of 0.00 mm/s, an effective magnetic field of 458 kOe, and a relative spectral area of 13%. The paramagnetic components are represented by three doublets. The first doublet has an isomer shift of 1.15 mm/s, quadrupole splitting of 2.75 mm/s, and a relative spectral area of 37%. The second doublet is characterized by an isomer shift of 0.34 mm/s, quadrupole splitting of 0.56 mm/s, and a relative spectral area of 37%. The third doublet shows an isomer shift of 0.44 mm/s, quadrupole splitting of 0.77 mm/s, and a relative spectral area of 8%. The spectrum of the magnetic part is a superposition of magnetically ordered and paramagnetic components (two sextets and one doublet). Two magnetically ordered sextets correspond to magnetite FeO·Fe2O3 or magnesium ferrite MgO·Fe2O3 with close parameters. The doublet has parameters close to those of iron hydroxide γ—FeOOH (lepidocrocite), with similar values for the non-magnetic part (doublet 1) The magnetically ordered states correspond to nearly stoichiometric magnetite (Fe3O4). The tetrahedral sublattice shows an isomer shift of 0.28 mm/s, quadrupole splitting of 0.01 mm/s, an effective magnetic field of 492 kOe, and a relative spectral area of 33%. The octahedral sublattice exhibits an isomer shift of 0.67 mm/s, quadrupole splitting of 0.01 mm/s, an effective magnetic field of 461 kOe, and a relative spectral area of 64%.
The particle size analysis (Figure 4b) concludes that size has a "negligible" effect on leaching efficiency, but the data show a ~2.7% variation (79.28–81.94%) across the tested range (0.071–1.25 mm). This variation may be statistically significant given the high magnesium content (~25–27% Mg) in AOW. The authors should perform a statistical analysis (e.g., ANOVA) to confirm the insignificance and discuss potential reasons (e.g., fiber morphology dominating over size) to avoid misleading conclusions.
Question 3.
The particle size analysis (Figure 4b) concludes that size has a "negligible" effect on leaching efficiency, but the data show a ~2.7% variation (79.28–81.94%) across the tested range (0.071–1.25 mm). This variation may be statistically significant given the high magnesium content (~25–27% Mg) in AOW. The authors should perform a statistical analysis (e.g., ANOVA) to confirm the insignificance and discuss potential reasons (e.g., fiber morphology dominating over size) to avoid misleading conclusions.
Response:
We thank the reviewer for the question. Based on the calculated correlation coefficients and their significance values, it follows that the particle size is not a significant factor in the final equation. As part of the multifactor analysis, we evaluated the contribution of each variable through multiple correlation coefficients and their statistical significance. For the parameter “particle size,” the nonlinear multiple correlation coefficient was found to be R = 0.07 with tR = 0.11. Since the tR value is much lower than 2, this parameter can be considered statistically insignificant.
Analysis of Particle Size as a Non-significant Factor in the Model:
The significance or insignificance of a partial function is determined using the nonlinear multiple correlation coefficient:
|
, |
The significance value tR at the 5% level, which is sufficient for chemical and metallurgical studies, calculated by the following formula:
where N is the number of measured points, K is the number of factors, Ye is the experimental value, Yt is the theoretical value, and Ya is the average experimental value. For the analysis of the significance of a partial function, N equals 5 and K equals 1.
The results of the correlation coefficient calculation and its significance are presented in Table 6. As can be seen from the data, four out of the five partial dependencies are statistically significant.
Table A2. Correlation coefficient R and statistical significance tr for the partial functions
|
Function |
R |
tr |
Function significance |
|
Y1 |
0.96 |
20.02 > 2 |
Significant |
|
Y2 |
0.99 |
90.31 >> 2 |
Significant |
|
Y3 |
1.00 |
1116.59 >> 2 |
Significant |
|
Y4 |
0.97 |
30.80 > 2 |
Significant |
|
Y5 |
0.07 |
0.11 < 2 |
Not significant |
Question 4.
The proposed closed-cycle process for HNO₃ regeneration via thermal decomposition of Mg(NO3)2 to MgO and NO2 is conceptually promising but lacks experimental validation. The manuscript would be strengthened by including preliminary data on the decomposition efficiency, NO₂ capture, and acid regeneration rates, even if from bench-scale tests. Without this, the claim of a "closed cycle" remains speculative and reduces the practical impact of the study.
Response:
We understand that the statement regarding the closed-loop nature of the process may appear speculative without experimental validation. However, the proposed approach is based on established industrial precedents and well-known chemical processes.
The decomposition of magnesium nitrate into magnesium oxide (MgO) and nitrogen oxides (NO2) through thermal treatment is a well-studied and industrially implemented operation, for instance, in hydrometallurgical processing of lateritic nickel ores (the so-called Direct Nickel Process) (https://doi.org/10.1016/j.mineng.2023.108170; https://altiliumgroup.com/the-dni-process/).
Studies confirm that magnesium nitrate decomposes at a relatively low temperature (~550 °C), which is significantly lower than that required for sulfate decomposition (~1200 °C). High decomposition efficiencies of magnesium nitrate (∼99.75%) are achieved under controlled conditions (https://doi.org/10.1080/09593330.2022.2121182; https://doi.org/10.1021/acs.iecr.5c02714.s001).
Therefore, the theoretical justification for obtaining market-grade MgO and gaseous NO2 is robust and grounded in well-established industrial chemical operations. The regeneration of nitric acid from gaseous NO2 is a standard industrial process forming the basis of the Ostwald process for HNO3 production.
Question 5.
The mathematical model for magnesium extraction (Section 3.2.3) uses orthogonal design and partial functions but does not adequately address potential interactions between factors (e.g., temperature and acid concentration). The model should be expanded to include interaction terms, and its predictive accuracy should be validated against an independent dataset not used in model fitting. Metrics such as R², RMSE, or confidence intervals for predictions should be provided to quantify model reliability?
Response:
We thank the reviewer for the valuable comment, which helped us clarify the mathematical part of the study. In the original work, we employed the Protodyakonov method, which is based on orthogonal matrices and a system of partial functions. This approach provides an independent assessment of the main effects of factors and allows determining their relative influence on the response with a minimal number of experiments. However, as correctly noted, this method does not explicitly account for cross-interactions between factors.
To quantitatively evaluate such interactions, we performed an additional second-order regression analysis using the same 25 experimental data points and applied a polynomial model including quadratic and cross terms. Thus, the Protodyakonov calculations were used as a baseline linear approximation, while the refined second-order model was applied to verify factor interactions and validate the predictive capability of the approach.
3.2.2 Design of Experiments
The Latin square planning method was applied in this study, and the Protodyakonov equation was used to construct the experimental model [40]. This approach allows for the evaluation of how multiple factors affect the magnesium leaching rate and facilitates the identification of the interrelationships among them. The model can also be employed to predict the process outcomes when the operating conditions are modified.
Based on the obtained data, a second-order regression model was developed to describe the relationship between process parameters and magnesium extraction. The model includes linear, quadratic, and cross terms that reflect both the main and combined effects of the studied factors. All variables were centered relative to their mean values, which ensured stable estimation of the interactions.
This experiment aimed to establish how the magnesium extraction process functionally depends on five selected parameters. The complete experimental matrix is presented in Table A1 in Appendix A. The factors chosen for the leaching experiments were nitric acid concentration (AC), liquid-to-solid ratio (L:S), leaching temperature (T), leaching time (t), and particle size (PS).
Reviewer 3 Report
Comments and Suggestions for Authors
The manuscript titled “Nitric Acid Leaching for Magnesium Extraction from Asbestos Ore Waste: From DoE to Predictive Modeling and Cost-Efficient Optimization” presents a systematic study on magnesium extraction using nitric acid and develops a cost-efficiency optimization model. The topic is relevant and timely, addressing both environmental and industrial aspects of asbestos waste utilization. The experimental design and modeling approach are appropriate, and the manuscript provides useful data for future scale-up studies.
However, a few issues need to be addressed before publication. I therefore recommend minor revision.
- The manuscript presents an interesting nitric-acid leaching model for magnesium recovery, but the validation section lacks comparison between experimental and predicted data (e.g., parity plots, residual analysis, or error metrics). Including such validation would strengthen confidence in the model’s predictive capability and industrial applicability.
- The text is generally understandable but contains grammatical and typographical errors (e.g., “open a pathway” → “opens a pathway”; “dependents” → “dependence”). A careful language edit is recommended to improve clarity.
- Ensure consistency in units (e.g., g L⁻¹ vs. g/L) and typographical presentation of variables such as Mg²⁺, °C, and molar quantities throughout the figures and text.
- Figures 3–6 could benefit from clearer axis labeling and higher-resolution images. In particular, color maps should include a scale bar and indicate the variable represented by color.
- The description of the reactor setup and leaching procedure (Section 3.2.1) should mention stirring speed and vessel volume, as these can influence mass-transfer kinetics.
- A few citations appear duplicated (e.g., Ref. 3 and Ref. 28). Please check for repetition and ensure DOI formatting and numbering follow Molecules journal guidelines.
Author Response
Question 1.
The manuscript presents an interesting nitric-acid leaching model for magnesium recovery, but the validation section lacks comparison between experimental and predicted data (e.g., parity plots, residual analysis, or error metrics). Including such validation would strengthen confidence in the model’s predictive capability and industrial applicability.
Response:
Model Validation and Error Analysis:
The verification of the generalized equation was done by comparing the calculated results with the experimental data. For this purpose, the values of the partial functions that correspond to the conditions of each of the 25 experiments were substituted into the equation. The calculated results Yp are shown in Table A3. The correlation coefficient R for N = 25 and K = 5 was 0.97, and the significance tR was 62.83, which confirms that the generalized equation is adequate. The error of the equation was calculated using the standard formula, and the total deviation σ was 4.39%.
,
Table A3 Accuracy Assessment of the Regression Model Based on Experimental Data
|
№ |
|||
|
1 |
36.47 |
43.15 |
6.68 |
|
2 |
85.87 |
81.61 |
4.26 |
|
3 |
62.32 |
65.16 |
2.84 |
|
4 |
98.70 |
102.77 |
4.07 |
|
5 |
96.23 |
94.27 |
1.96 |
|
6 |
68.67 |
73.43 |
4.76 |
|
7 |
84.19 |
85.57 |
1.38 |
|
8 |
100.42 |
102.15 |
1.73 |
|
9 |
83.03 |
81.97 |
1.06 |
|
10 |
58.08 |
59.17 |
1.09 |
|
11 |
71.69 |
70.22 |
1.47 |
|
12 |
81.04 |
81.23 |
0.19 |
|
13 |
89.35 |
90.31 |
0.96 |
|
14 |
52.06 |
53.44 |
1.38 |
|
15 |
96.64 |
95.83 |
0.81 |
|
16 |
97.95 |
92.98 |
4.97 |
|
17 |
97.84 |
93.87 |
3.97 |
|
18 |
76.19 |
70.51 |
5.68 |
|
19 |
92.91 |
99.33 |
6.42 |
|
20 |
64.92 |
69.84 |
4.92 |
|
21 |
96.56 |
99.55 |
2.99 |
|
22 |
65.40 |
65.07 |
0.33 |
|
23 |
78.76 |
77.26 |
1.50 |
|
24 |
87.55 |
78.11 |
9.44 |
|
25 |
93.46 |
93.49 |
0.03 |
Question 2
The description of the reactor setup and leaching procedure (Section 3.2.1) should mention stirring speed and vessel volume, as these can influence mass-transfer kinetics.
Response:
Thank you for the comment. We have added the necessary data on the stirring speed and vessel volume in Section 3.2.1, as these parameters can influence mass-transfer kinetics.
3.2. Metods
3.2.1 Acid Leaching
The agitation leaching experiments were carried out in sealed reactor equipped with a gas-release valve. Prior to leaching, the AOW was heated in a muffle furnace at 100 °C overnight to remove any residual liquid phase.
Afterward, the asbestos ore waste was ground in a laboratory disk grinder to obtain a uniform powder. The grinding process was carried out in two stages. The first was a rough stage with a gap of 0.35 mm and a duration of 3 minutes. The second was a fine stage with a gap of 0.18 mm and a duration of 3 minutes, with manual mixing every 2 minutes. The rotation speed was 875 rpm, and the batch load was 100 grams. The temperature of the grinder housing was kept below 50 degrees Celsius to avoid moisture loss from crystal hydrates. After each batch, the disks were cleaned with compressed air and ethanol.
Magnetic separation was carried out using a laboratory roll-type separator with magnetic susceptibility in the range of 10-6 to 10-7 m3/kg. The working gap between the poles was 2 millimeters, the magnetic field induction 0.7 to 1.0 T, the roller rotation speed was 0.5 m/s, and the chute inclination angle was 20 degrees. The process was performed under dry conditions, and the material passed through the separator three times to ensure efficient separation. A sample weight of 15 g was used for each experiment. The reactor temperature was controlled by a LOIP LT-108A thermostat that was connected to the leaching reactor. The reactor was placed on a LOIP LS-110 shaker, and stirring was performed using the rolling method in the horizontal plane. The reactor volume was 200 ml, and the shaking speed was 200 RPM. After leaching, the samples were filtered and washed with distilled water until a neutral pH was reached.
Reviewer 4 Report
Comments and Suggestions for Authors
The reviewer agrees with the importance of the treatment of asbestos ore wastes. This paper considers the optimal processing conditions from the aspects of including cost effectiveness. However, the following points should be considered for the clarity of the readers.
- Please add the mol description of the calculation of the required amount of HNO3.
- The authors estimated the required energy by calculation. This is a fundamental procedure, however, energy loss of the machines should also be considered in the actual processing. The reviewer recommends checking the actual energy consumption using a Watt monitor or other procedure, and comparing it with the calculation results, if the authors can.
- The effect of the particle size is important because of the low energy efficiency of grinding. As long as the results of this paper, fine pulverization is not required for the effective treatment and this is good news for the operator. The reviewer understands that the authors excluded this factor, however, please add this cost for a more appropriate cost evaluation if the authors can.
There are too many mistakes as follows, including those other than English quality. Please check the script before submission.
- Figure a → Figure 4. a? (L. 207)
- Figure 7 → Figure 6? (L. 264)
- Figure 8 → Figure 7? (L. 270)
- Metods → Methods (L. 280)
- Lack of ref. number (L. 358)
- 5. Conclusions → 4. Conclusions (L. 383)
- Superscript and subscript (throughout this paper)
Author Response
Question 1.
Please add the mol description of the calculation of the required amount of HNO3.
Response:
Thank you for the comment and the suggestion to add a molar description of the required HNO₃ amount. The composition of asbestos ore waste (AOW) depends strongly on the deposit and may vary even within the same deposit. Therefore, instead of using a fixed “typical” composition, we estimated the required amount of nitric acid from the actual elemental analysis data for our specific AOW sample.
To estimate the minimum amount of nitric acid required to leach the Mg from asbestos ore was calculated based on mineralogical and elemental compositions. Based on this calculation, the minimum acid concentration was defined. The leaching process is expected to proceed through the following reactions:
Question 2.
The authors estimated the required energy by calculation. This is a fundamental procedure, however, energy loss of the machines should also be considered in the actual processing. The reviewer recommends checking the actual energy consumption using a Watt monitor or other procedure, and comparing it with the calculation results, if the authors can.
Response:
We thank the reviewer for the constructive comment. In the calculations presented in the paper, the energy consumption was evaluated theoretically, based on the stoichiometric and thermodynamic parameters of the processes (heat capacity of the reaction mass, heating temperature, mass of reagents, and heat transfer through the reactor walls). This assessment made it possible to determine the minimum energy required for the process under idealized conditions, which corresponds to the commonly accepted approach for preliminary techno-economic analysis.
Question 3.
The effect of the particle size is important because of the low energy efficiency of grinding. As long as the results of this paper, fine pulverization is not required for the effective treatment and this is good news for the operator. The reviewer understands that the authors excluded this factor, however, please add this cost for a more appropriate cost evaluation if the authors can.
Response:
We thank the reviewer for this valuable comment. In the present study, the processed raw material represents asbestos fiber production waste (AOW), which has already undergone crushing, grinding, and fiber separation stages within the enterprise’s technological scheme. During the fiber extraction from serpentine, the mineral mass is repeatedly subjected to mechanical treatment, leading to partial matrix destruction and the formation of fine powder fractions. Therefore, the resulting waste already possesses the required granulometric characteristics, and no additional grinding is needed prior to acid leaching.
Accordingly, within the proposed technological scenario for recycling existing asbestos production waste, the energy consumption for grinding was not included in the economic model, since this operation has effectively already been performed during the fiber extraction process and is not part of the proposed recycling cycle.
At the same time, if the enterprise were to be reoriented from asbestos fiber production to the direct production of magnesium oxide from serpentine—without the intermediate fiber extraction stage—the energy and economic model should additionally include:
- energy costs for drilling and blasting operations and ore extraction;
- transportation of raw material from the quarry to the processing site;
- crushing and grinding to the required particle size;
- and energy consumption for raw material classification.
Such an assessment can be performed using data on energy consumption from mining and beneficiation industries (https://doi.org/10.1016/C2010-0-65478-2). In this case, the share of energy consumption associated with extraction and grinding may account for up to 25–35% of the total production cost of MgO, reflecting the need to revise the economic estimations when transitioning to a full technological cycle.
Round 2
Reviewer 1 Report
Comments and Suggestions for Authors
Dear Authors,
The manuscript has been improved based on my feedback. I recommend the paper for publication.
Regards
Author Response
Dear reviewer,
We sincerely thank you for your positive evaluation and recommendation for publication. We truly appreciate your valuable comments, which helped us to improve the overall quality of the manuscript. In the final version, we have also enhanced the quality of several figures and tables to ensure clearer visualization and better presentation of the results.
With kind regards,
The Authors
Reviewer 2 Report
Comments and Suggestions for Authors The authors have diligently addressed several reviewer comments in their revised manuscript. However, two minor issues require further attention to meet Molecules's rigorous standards for mechanistic insight and data reproducibility. 1. While the regression model demonstrates strong predictive power (R² = 0.976), the manuscript would benefit from a more comprehensive discussion of model validation and error analysis beyond the provided RMSE values. Specifically, the authors should include a detailed residual analysis to verify the homoscedasticity and normal distribution of errors, which is critical for ensuring the model's reliability across the entire parameter space. Additionally, the discussion should explicitly address potential limitations in extrapolating the model to industrial-scale operations, particularly regarding mixing dynamics and heat transfer effects that may differ from laboratory-scale reactors, as these factors could influence the practical application of the proposed cost-optimization framework without requiring new experimental data. 2. The cost-efficiency analysis in Section 3.2.4, while comprehensive, lacks transparency in several key assumptions that underpin the economic model. The authors should provide greater methodological detail on how corrosion costs were quantified, specifically clarifying the corrosion rate equations and their dependence on acid concentration and temperature, as this nonlinear relationship significantly impacts the optimum identified in Figure 6. Furthermore, the manuscript should explicitly justify the selection of the 13% Cr steel corrosion data and explain how these values were adapted to the specific leaching conditions, as this would strengthen the credibility of the economic conclusions and ensure reproducibility without necessitating additional experiments.Author Response
Question 1.
- While the regression model demonstrates strong predictive power (R² = 0.976), the manuscript would benefit from a more comprehensive discussion of model validation and error analysis beyond the provided RMSE values. Specifically, the authors should include a detailed residual analysis to verify the homoscedasticity and normal distribution of errors, which is critical for ensuring the model's reliability across the entire parameter space. Additionally, the discussion should explicitly address potential limitations in extrapolating the model to industrial-scale operations, particularly regarding mixing dynamics and heat transfer effects that may differ from laboratory-scale reactors, as these factors could influence the practical application of the proposed cost-optimization framework without requiring new experimental data.
Response:
The authors would like to thank the reviewer for the careful examination of the revised manuscript and for the constructive comments aimed at improving the statistical rigor and methodological transparency of the presented work. The reviewer’s attention to the validation of the model and the detailed economic analysis, particularly regarding the nonlinear corrosion costs, is highly valuable and essential for ensuring the practical applicability of the results under industrial conditions.
To confirm the reliability of the regression analysis, including the significance of coefficients and interaction terms (for example, between acid concentration and temperature), two main assumptions were examined: the normal distribution of errors and homoscedasticity, which means constant variance of errors. According to Table A3 in the Appendix, the residuals were obtained as the difference between the experimentally measured magnesium yield and the value predicted by the generalized equation for all 25 experiments. The average value of residuals is close to zero, which indicates that the model is adequate and correctly constructed using the least squares method.
A detailed consideration of the residual distribution based on the accuracy check of the model shows that most deviations are moderate. In 23 out of 25 experiments, the absolute deviation is less than 6.7%. The largest deviation, 9.44%, was found in experiment No. 24, where the predicted yield was lower than the experimental one (87.55% versus 78.11%). This experiment was performed at a moderate temperature (65 °C) but with high acid concentration (525 g/L) and high solid-to-liquid ratio (7:1). Such a deviation may appear due to small-scale mass-transfer limitations or minor nonidealities near the borders of the studied range, which are not always easy to describe in the generalized model. Nevertheless, since the overall residual distribution does not show systematic bias and is close to normal, the statistical reliability of the model and its coefficients can be considered satisfactory.
The assumption of homoscedasticity ensures that the model keeps the same predictive accuracy for all predicted values. Violation of this condition usually results in a visible increase or decrease of residual variance with changing predicted yield. The analysis of the residuals as a function of the predicted values demonstrates no systematic pattern. The residuals are evenly distributed through the whole range of magnesium yields. Therefore, the model can predict results with similar reliability both in the region of low yields (at low concentrations and temperatures) and in the region of high, near-optimal yields. This provides a solid basis for further evaluation of the process efficiency.
The authors understand that the proposed model was developed and tested under laboratory scale conditions. The statistical analysis confirms that it is adequate within this range. At the same time, we recognize that direct transfer of the results to industrial systems should be made carefully. In industrial practice, several factors can behave differently, especially those related to mass transfer. In our laboratory experiments, the particle size did not show statistical significance. This means that under laboratory conditions the reaction kinetics play the main role, and the process is not limited by mass transfer. When the process is scaled up to industrial reactors of larger volume, and when high solid to liquid ratios is used for economic reasons, the mass transfer effect can become more important.
In such cases the predictive accuracy of a model based mainly on kinetics may be reduced. This can lead to an overestimation of the magnesium yield. Therefore, at an industrial scale the particle size factor may again become significant. It may require finer grinding of the raw material or stronger mixing of the suspension. Both of these factors can increase the energy consumption and influence the Cenergy term in the economic evaluation.
Question 2.
- The cost-efficiency analysis in Section 3.2.4, while comprehensive, lacks transparency in several key assumptions that underpin the economic model. The authors should provide greater methodological detail on how corrosion costs were quantified, specifically clarifying the corrosion rate equations and their dependence on acid concentration and temperature, as this nonlinear relationship significantly impacts the optimum identified in Figure 6. Furthermore, the manuscript should explicitly justify the selection of the 13% Cr steel corrosion data and explain how these values were adapted to the specific leaching conditions, as this would strengthen the credibility of the economic conclusions and ensure reproducibility without necessitating additional experiments.
Response:
We sincerely appreciate the reviewer’s constructive comment regarding the transparency of the corrosion cost estimation and the justification for using 13 % Cr steel. In the revised manuscript, additional methodological details have been added to clarify how the corrosion costs were quantified and how the selected data were integrated into the cost model.
The 13 % Cr martensitic stainless steel was selected as a reference material in order to evaluate the influence of corrosion on the total process cost. This alloy was chosen because it exhibits a higher corrosion rate compared with more corrosion-resistant grades such as AISI 304L, and therefore allows the estimation of the possible maintenance and equipment replacement costs within the overall economic analysis.
The corrosion cost (Ccorr) was incorporated into the total cost model to account for potential material losses during nitric acid leaching. Because the primary aim of this work was to evaluate the overall cost-efficiency of the process, a practical and simplified estimation method was applied instead of a detailed corrosion model.
Corrosion rate data for 13 % Cr martensitic stainless steel were taken from the literature,
where comparable acid concentrations and temperatures were reported. For each experimental condition, the corresponding corrosion rate value was directly used and converted into a cost term using a coefficient (Kcorr) that reflects the relative financial impact of equipment degradation:
Ccorr = rcorr * Kcorr
This approach provides a consistent and transparent estimation suitable for coarse-scale techno-economic analysis, where corrosion contributes only as a secondary cost factor. Although 13 % Cr steel exhibits limited corrosion resistance in strongly oxidizing media such as nitric acid, it was adopted here as a reference material to ensure comparability of cost estimates under different experimental conditions.
